# Advances in Ophthalmic Optogenetics: Approaches and Applications

**DOI:** 10.3390/biom12020269

**Published:** 2022-02-08

**Authors:** Philipp P. Prosseda, Matthew Tran, Tia Kowal, Biao Wang, Yang Sun

**Affiliations:** 1Department of Ophthalmology, Stanford University School of Medicine, 1651 Page Mill Road, Palo Alto, CA 94305, USA; philipp.p.prosseda@gmail.com (P.P.P.); matthewtran@med.unr.edu (M.T.); tjkowal@stanford.edu (T.K.); wangbiao19871007@gmail.com (B.W.); 2Reno School of Medicine, University of Nevada, Reno, NV 89557, USA; 3Palo Alto Veterans Administration, Palo Alto, CA 94304, USA

**Keywords:** retinitis pigmentosa, optogenetics, glaucoma, degenerative eye diseases

## Abstract

Recent advances in optogenetics hold promise for vision restoration in degenerative eye diseases. Optogenetics refers to techniques that use light to control the cellular activity of targeted cells. Although optogenetics is a relatively new technology, multiple therapeutic options are already being explored in pre-clinical and phase I/II clinical trials with the aim of developing novel, safe, and effective treatments for major blinding eye diseases, such as glaucoma and retinitis pigmentosa. Optogenetic approaches to visual restoration are primarily aimed at replacing lost or dysfunctional photoreceptors by inserting light-sensitive proteins into downstream retinal neurons that have no intrinsic light sensitivity. Such approaches are attractive because they are agnostic to the genetic causes of retinal degeneration, which raises hopes that all forms of retinal dystrophic and degenerative diseases could become treatable. Optogenetic strategies can also have a far-reaching impact on translational research by serving as important tools to study the pathogenesis of retinal degeneration and to identify clinically relevant therapeutic targets. For example, the CRY-CIBN optogenetic system has been recently applied to animal models of glaucoma, suggesting a potential role of OCRL in the regulation of intraocular pressure in trabecular meshwork. As optogenetic strategies are being intensely investigated, it appears crucial to consider the opportunities and challenges such therapies may offer. Here, we review the more recent promising optogenetic molecules, vectors, and applications of optogenetics for the treatment of retinal degeneration and glaucoma. We also summarize the preliminary results of ongoing clinical trials for visual restoration.

## 1. Introduction

Optogenetic molecules can be delivered and targeted to different regions of the mouse visual system. Once in place, optogenetic modules can be activated using light illumination at the appropriate wavelength for the respective module used (Figure 1).

Optogenetic therapy, a new treatment strategy that combines optic and genetic techniques, offers fresh hope for targeting various eye conditions, regardless of the gene responsible for the disease. Optogenetics is a relatively novel and elegant biological technique based on genetic engineering technologies, which makes use of light to control the localization of proteins of interest with a high temporal and spatial resolution to target specific areas within cells, tissues, or organs [1]. Inherited retinal degenerations are estimated to affect 1 in 4000 people worldwide and have become the most common cause of blindness amongst the working-age population in developed countries [2,3,4].

Over the past decade, optogenetics has been intensively applied in both groundbreaking experimental laboratory settings for the discovery of novel biological mechanisms and in the explorations of novel technological techniques to convert optogenetics into innovative therapeutic methods [5,6,7,8]. Especially from a therapeutic standpoint, the feasibility of this technique was proven on multiple occasions [9,10]. One recent example demonstrated that optogenetic molecular tools can be successfully adapted to restore neuron light sensitivity by targeting gene expression in selective retinal cell types. In rodent models of retinitis pigmentosa (RP), a group of progressive, hereditary diseases of the retina that lead to incurable blindness, light-insensitive retinas were successfully reactivated using optogenetics [11]. Importantly, these preliminary studies showed that for the reactivation to occur, it sufficed to deliver genes of light-activated proteins to the few surviving cell types in the remaining retinal circuits, supporting the advantage of optogenetics over classical gene therapy, which relies on the presence of the original cell population [11].

Although in recent years, novel approaches to treat diverse eye conditions have emerged from the field of gene therapy, most of these approaches are based on classical local gene-replacement techniques, accompanied by their typical problems and limitations [12,13,14]. There are two main disadvantages of classical gene therapy, which can be summarized by the example of RP. The most obvious limitation is that gene therapy can only target and restore the function of genes in living cells, while the genetic defects in RP result in progressive dysfunction and irreversible loss of photoreceptors [15]. The photoreceptors cells are of two types—rods and cones. These specialized cells, located in the outer neural retina, contain the visual pigments (rod and cone opsins) and function in the capture of light and its conversion into electrical signals via phototransduction [16]. Rods are highly sensitive to light and play a role in peripheral vision and in vision under low-light conditions. Cones function under bright light and are responsible for central discriminative vision and for color detection. In the healthy retina, photoreceptors form synaptic connections with bipolar cells, which in turn synapse with retinal ganglion cells (RGCs). These connections are referred to as the ‘primary vertical pathway’ to the brain, whereas amacrine cells and horizontal cells form the lateral connections [17]. Indeed, photoreceptor cell death tends to be the final, irreversible event in many blinding diseases, including RP [18]. Despite this, the remaining visual pathway remains largely viable and provides the rationale behind the methods as well as an appreciation of the directions in which the field of gene therapy is headed, including optogenetics [19,20,21]. However, even if gene therapy could be applied early, before the loss of rod photoreceptor cells becomes too incapacitating, RP is caused by the mutation of not one but of up to 50 different genes, making the targeting choice for gene therapy a daunting undertaking [22,23,24]. Even so, except for LUXTURNA, a gene replacement therapy for the rare form of autosomal recessive RP caused by mutations in the retinal pigment epithelium 65 (*RPE65*) gene, there is no approved therapy for RP [12]. Gene-replacement therapy primarily aims to prevent vision loss by slowing the rate of progression of photoreceptor degeneration, which makes it unsuitable for patients with undetermined genotypes and those who present with late-stage disease in which there is profound photoreceptor cell loss. In such degenerated retinas, downstream neurons can be transduced to express opsins thereby conferring sensitivity to light [9,10]. Such strategies are attractive because they have the potential to salvage vision irrespective of the genetic etiology. Hence, optogenetic approaches to treat diseases that affect the inner retina, such as glaucoma, are also being explored [5,6]. As optogenetic strategies are being intensely investigated, it appears crucial to consider the opportunities and challenges such therapies may offer. Here, we review the more recent promising optogenetic molecules, vectors, and applications of optogenetics for the treatment of retinal degeneration and glaucoma. We also summarize the preliminary results of ongoing clinical trials for vision restoration.

## 2. Opsins Used for Optogenetic Therapy

The opsin genes used for optogenetic vision restoration are divided into two superfamilies: microbial opsins (Type 1) and animal opsins (Type 2). Although both opsin families encode photoactive proteins consisting of seven transmembrane α-helical domains, they greatly differ in their functions, sensitivity to light, and utility for vision restoration [25]. In Type 1 opsins, the proteins are covalently linked to the all-*trans*-retinal chromophore, which isomerizes upon light absorption to induce a conformational change in the opsin and directly influence ion channels or pumps. By contrast, Type 2 opsins are usually covalently bound to 11-*cis*-retinal, and light illumination activates downstream intracellular G-protein-coupled receptor (GPCR) signaling cascades to indirectly influence ion channels [26]. For this reason, microbial opsins, such as channelrhodopsin-2 (ChR2), are the most widely used optogenetics tools in neuroscience to impart light-induced membrane permeability to neurons [27]. For applications specific to ophthalmology, four opsin classes have proven useful in optogenetic vision restoration: channelrhodopsin (ChR), halorhodopsin (HR), melanopsin (OPN4), and human rhodopsin (RHO) [26,28,29].

The different functional properties of each opsin, such as ion selectivity and light sensitivity, enable researchers to optimally target specific subpopulations of neurons within the retina. For example, light-insensitive cone photoreceptors with damaged outer segments caused by retinal degenerative diseases, such as RP, can be reactivated using HR. HR derived from the archaeon *Natronomonas pharaonis* (NpHR) acts as a light-gated chloride pump with a peak wavelength sensitivity of 580 nm; light activation leads to hyperpolarization of the cell, mirroring the native response of a cone [30]. Alternatively, optogenetics can be used to target the surviving inner retinal neurons following photoreceptor loss, such as retinal bipolar cells via ectopic expression of ChR2 or RHO [29,31].

The ChRs are a family of cation channel-forming opsins that were the first to be identified as optogenetics tools. ChR2, isolated from the green algae *Chlamydomonas*, is the most commonly used microbial opsin to date and maximally absorbs blue light at a wavelength of approximately 470 nm, which generates a depolarizing inward current in the cell [32]. Several studies have successfully exploited the ectopic expression ChR2 in retinal bipolar cells to rescue visual function in the rd1 mouse model of photoreceptor degeneration [33,34]. ChR2 expression in ON bipolar cells mediated by intravitreal injection of a recombinant adeno-associated virus (AAV) vector leads to the restoration of ON and OFF visual responses at the retinal and cortical levels, highlighting its potential for optogenetic therapy. Natural and engineered variants of ChR, such as ChrimsonR and ChronosFP, have different kinetics and sensitivities to light and are also used in optogenetic therapies [35].

RHO is the native visual pigment in rod photoreceptors; its ectopic expression in ON bipolar cells effectively restores visual responses to light in rd1 mice [36]. In fact, light responses of RHO-treated mice are more sensitive and occur at lower light intensities than those of ChR2-treated mice [37]. Human opsins such as RHO and OPN4 are GPCRs, which have integral signal amplification cascades in which a single photon can activate multiple downstream signaling proteins that gate cyclic nucleotide-gated ion channels. In addition to their signal amplification, the ectopic expression of endogenous human proteins may be preferable to that of other species for clinical applications to limit the risk for non-self immune responses [29,37].

In cases of advanced retinal degeneration, RGCs can be targeted with OPN4 or ChR2 [28,38]. OPN4 is a mammalian opsin expressed in a subpopulation of photoreceptive ganglion cells, which support a variety of non-image-forming visual functions, such as circadian photoentrainment and the pupillary light reflex, and are categorized as intrinsically photosensitive retinal ganglion cells (ipRGCs) [28]. Upon photoactivation, OPN4 acts as a GPCR to open cation channels in the cell membrane of ipRGCs. [39]. OPN4 has been successfully expressed in the RGCs of rd1 mice; these cells show greater light sensitivity, yet with a slower and more sustained response than RGCs transduced with ChR2 [26,38,39].

## 3. Cryptochrome-Based Dimerizers in Optogenetics

An alternative branch of optogenetics derives from different families of blue light receptors in higher plants, fungi, and algae. Studies from this branch of optogenetics have principally focused on photosensory receptors, which, in contrast to photosynthetic pigments responsible for photosynthesis, mediate non-photosynthetic light responses. These types of light photolyase-like receptors were originally discovered in *Arabidopsis thaliana*, a plant in which the genome encodes multiple different types of photoreceptors; these include phototropins (phot1 and phot2), three LOV/F-box/Kelch domain proteins (ZTL, FKF, and LKP2), red-light receptor phytochromes (phyA, phyB, phyC, phyD, and phyE), and the blue-light receptor photolyase-like flavoproteins commonly known as cryptochromes (CRY1 and CRY2) [40].

The more extensively studied cryptochromes mediate blue-light regulation of gene expression and photomorphogenic responses not only in plants but also in other organisms. As a consequence, the cryptic (CRY) origin of this protein resulted in the overall naming of these optogenetic modules. Blue-light receptors have been extensively used in experimental optogenetic approaches. These wide-ranging approaches include interactions between photoreceptor cryptochrome 2 (CRY2) and the putative transcription factor cryptochrome-interacting basic-helix-loop-helix 1 (CIB1), and various interactions involving light-oxygen-voltage-sensing domains (LOV domains), which are photosensors found in bacteria, archaea, plants, and fungi that detect blue light via a flavin cofactor [41,42].

Similar to cryptochromes, LOV domains can be found at the N-terminal of diverse signaling and regulatory domains, such as sensor histidine kinases, DNA-binding domains, and factor regulators [40,42]. Structurally, LOV domains are defined by the presence of an active site flavin cofactor and a GXNCRFLQ consensus flavin-adduct protein motif [43,44]. Upon blue-light absorption, these flavin-binding domains function in the regulation of enzymatic activity and signaling cascades that govern cellular responses, such as plant phototropism and bacterial phototaxis [45,46].

## 4. Approaches for the Delivery of Optogenetic Therapy in the Eye

The successful delivery and expression of various optogenetic constructs in chosen target areas represent a novel field of increasing interest to the scientific community, and multiple studies aim to optimize the expression and tissue specificity with which optogenetic modules target cells. AAV-based vectors, which have been studied extensively in animal models, are one of the favorite modes of expression. The final transduction efficiency of these vectors has proven superior to that of lentiviral vectors [47,48]. They have been shown to efficaciously express the light-sensitive opsin molecule or CRY2/CIBN-based enzymes in different parts of the eye [5,49]. Furthermore, the AAV-based delivery strategy has translated successfully into an effective treatment for patients; the first approved human AAV gene therapy (LUXTURNA) recently received approval for the treatment of an inherited retinal disease, Leber’s congenital amaurosis (LCA). This demonstrated clinical utility bolsters the choice of AAVs as a preferred delivery method for optogenetic approaches.

Although systemic delivery of AAV vectors in gene therapy has proven to be more challenging, AAV vectors in optogenetic applications have the advantage of offering flexibility in the choice of the delivery route [50]. Both intravitreal and subretinal injections provide a more targeted approach than systemic administration, and recent studies highlight the generally minimal adverse events in clinical trials [51]. Common drawbacks that have plagued the usage of AAVs as vectors for gene delivery, including the ability to only transfect particular cell types and a size limitation of packaging capability, do not seem to limit the applicability of optogenetic approaches because optogenetic coding sequences are relatively small in comparison to genes in supplementation strategies [52]. Moreover, transfection efficiency is likely to increase in the future since optogenetic therapies have recently driven investigation of novel engineered capsid variants able to specifically target a variety of different eye cell types, including the retinal pigment epithelium and trabecular meshwork [5,7].

Non-viral vectors, such as nanoparticles, for the ocular delivery of therapeutic materials to the retina are of particular interest for their advantages over viral-based strategies. Viral vectors are used to deliver functional genes to the retina, whereas nanoparticles can deliver both drugs and genes to the retina. Furthermore, the maximum capacity of AAVs to accommodate genomic information is approximately 4.8 kb, which prevents their use for the delivery of large genes, such as *USH2A* [53]. Several pre-clinical studies have used murine models to assess the safety and expression profiles of nanoparticle-mediated gene transfer to treat various inherited retinal diseases [32,36,54,55]. Liposome-protamine-DNA nanoparticles have been shown to promote cell-specific delivery and long-term expression of the *RPE65* gene in *RPE65* knockout mice in vivo [25]. Subretinal injection of compacted DNA nanoparticles containing the retinal degeneration slow (*Rds*) gene resulted in improved cone and rod function in an RP mouse model [26]. Further, biodegradable poly(β-amino ester) nanoparticle-based delivery of a plasmid encoding vascular endothelial growth factor (VEGF) neutralizing protein, p3sF1t1Fc, significantly suppressed vascular leakage and neovascularization in a mouse model of wet age-related macular degeneration (AMD) [36]. These studies suggest that nanoparticles are less immunogenic than viral therapies and can drive high levels of transgene expression in ocular tissues.

The next decade will likely see dramatic improvements in optogenetic delivery approaches thanks to additional natural and synthetic AAV serotypes and enhanced specificity of promoters that will restrict expression to specific cell types. Furthermore, the optogenetic modules themselves will improve because of developments in their sensitivity as artificial photoreceptors or as sensors responsive to a wide range of wavelengths, including near-infrared wavelengths [56].

## 5. Using Optogenetics in Retinal Degeneration and Glaucoma

Several light-induced molecules have been applied to eye diseases, particularly in the field of retinal degeneration. Retinal degeneration primarily affects light-detecting rod cells, which are mainly responsible for peripheral and dim light vision, and cone photoreceptors, which function most effectively in relatively bright light and are primarily responsible for high acuity central vision and color vision. Retinal degenerations cause severe visual impairment and blindness in millions of individuals worldwide [57,58,59]. Current research focuses on investigating the role of each gene in initiating and driving disease progression because there is a strong genetic component to the development of retinal degeneration, and over 300 causative genes have been identified to date [23,60,61]. This effort will eventually narrow down possible targets for gene replacement therapy via AAV. An optogenetic approach takes advantage of the evidence that, although the underlying role of the responsible genes is variable, the pathogenesis of retinal degeneration appears to converge on common final pathways, which produces considerable similarities in the physiological changes ultimately observed in the affected retina. These points of convergence can then be universally targeted by providing light-sensitive molecules that facilitate light perception in the surviving layers of cells in the neural retina [62].

The recent advances in optogenetics have also been applied to the field of glaucoma [5,6,63]. Glaucoma comprises a group of progressive optic neuropathies characterized by an abnormal increase in intraocular pressure (IOP), which damages the optic nerve over time and consequently results in injury and eventual loss of RGCs and retinal nerve fiber layers [64]. RGCs and their projections to the brain play a key role in visual perception; the loss of these cells ultimately represents the final stages of the converging pathways and mechanisms leading to visual impairment. The pressure for new therapeutic strategies has increased in recent years because elevated IOP is a leading risk factor for glaucoma [65,66].

Optogenetic approaches are used in animal models of glaucoma to understand the mechanisms that ultimately lead to the loss of RGCs. These studies use the optogenetic CRY2-CIBN system to study the role of a specific enzyme, inositol polyphosphate 5-phosphatase (OCRL), in modulating IOP prior to RGC loss [5,6]. Evidence of the role of this enzyme in glaucoma pathogenesis derives from patients with mutations in OCRL who develop Lowe syndrome, a multisystem disease causing abnormalities in the brain, kidneys, and eyes. The development of high IOP characterizes the disease; thus, it provides an apt model for understanding congenital glaucoma. Available evidence suggests that OCRL plays a key role in the mechanisms that modulate IOP early in the disease [5]. Interestingly, OCRL is localized in the primary cilia and plasma membrane of trabecular meshwork (TM) cells, a spongy tissue located around the base of the cornea involved in draining the aqueous humor via the anterior chamber of the eye. Targeted optogenetic modulation of OCRL to the primary cilia and plasma membrane was shown to cause contraction of the TM and an increase in outflow facility, which are correlated with a decrease in IOP [6]. These findings support the idea that defects in the regulation of aqueous humor outflow may be critical for the subsequent development of glaucoma.

Optogenetic approaches have also recently been applied to understanding the final stages of glaucoma. Under normal conditions, nerve cells that die in the eye because of glaucoma do not grow back, indicating that the loss of vision is inevitable and irreversible. For this reason, there has been a considerable effort to identify molecular mechanisms that will allow damaged RGCs to reconstitute nerve connections with the brain before dying [67]. One of these studies successfully applied a modified optical coherence tomography (OCT) technology to image the neuronal connections of genetically engineered animals with acute nerve cell injury in the retina. This specific platform, which allows direct imaging of neurons in living experimental animals, provides an extremely useful model for studying the correlation between the loss of specific neuron process (axon) connections and the death of the neurons, particularly when combined with the optogenetic studies designed to restore RGC growth mentioned above [67].

In this context, the current classical approach to glaucoma focuses on two main categories of research: mechanisms controlling intraocular pressure, which is a major risk factor for glaucoma, and mechanisms contributing to the development of neuroprotective therapies to protect RGCs. To expand the treatment possibilities in the later stages of the diseases, it would be desirable to add a third research category, which would stimulate lost RGCs to grow back or completely replace them by optogenetic modules that assume some of the functions of image transmission.

Importantly, if the above-mentioned research category is to be successful, the type of stimulation needed to restore original RGC function will not only require the identification of the responsible pathways but also their precise localization where these mechanisms are activated. It is, therefore, relevant that several studies provide tentative evidence that RGC survival depends in part on support from the target neurons to which they ultimately project, suggesting that targeting mechanisms should not exclusively focus on RGCs. However, this broader focus has important new implications for applying replacement signaling molecules to different cell types because the signaling pathways induced by target-derived factors are distinctly different from those activated when the same molecules are provided at the RGC [63]. The importance of this difference in signaling pathways receives support from a recent study showing that an optogenetic approach could be employed to induce a controllable, prolonged activation of neuronal activity in a visual target center in the brain. This approach enhanced RGC survival in a mouse glaucoma model and consistent neuronal activation by repeated stimulations. Specifically, when optogenetic constructs known as stabilized step function opsins (SSFO) were directed to the superior colliculus (SC), one of the targets of RGC axons in the mouse brain, they produced kinetics that allowed a lasting signal in response to only a brief light pulse. These results support the idea that increasing the neuronal activity of the visual system target centers involved in the bidirectional circuit between RGCs and the brain will protect against RGC degeneration over time and can potentially be exploited as a future strategy to increase neuroprotection in glaucoma [63].

A key consideration before using optogenetics to treat retinal degeneration in humans is safety. Determining the parameters of light stimulation necessary to drive the optogenetic proteins is of particular importance, as the retina can be damaged by certain light intensities and wavelengths [68,69]. Standard limits for ocular exposure to radiation have been defined, and the maximal permissible exposure for a given optogenetic therapy is determined by a variety of factors including wavelength, exposure duration, and pupil size [70,71,72]. Several pre-clinical optogenetic studies have used mouse and primate models to assess the safety and functional characteristics of opsins in terms of their translatability to human subjects [9,33,34,73]. These studies have demonstrated that while microbial opsins, such as ChR2, can restore visual responses when targeted to inner retinal neurons, the short wavelength of blue light sufficient to activate ChR2 requires an intensity that exceeds the safety threshold for retinal illumination in humans, increasing the risk of photochemical damage to the retina [74,75,76]. As a result, in the last few years, human and animal studies have favored the use of opsins with red-shifted wavelengths, such as ChrimsonR, which can be activated at light intensities well below the safety threshold [8,77,78].

Inflammation is another crucial factor to consider when evaluating the translational potential of optogenetic studies. The introduction and expression of foreign proteins, mainly of microbial origin, on the membranes of retinal cells pose an inherent risk for an undesired immune response. Furthermore, vectors derived from AAV are the most commonly used platform for delivery of retinal gene therapy and have been previously associated with inflammation in many tissues [79,80,81]. One general concern for AAV vectors is that pre-existing immunity and subsequent induced adaptive immunity following vector administration can significantly reduce retinal gene expression, as shown in pre-clinical studies [82]. Furthermore, there is growing appreciation for the risk of gene therapy associated uveitis following AAV administration, which appears to be related to the vector dose and route of administration [83,84]. However, optogenetic clinical trials thus far have demonstrated that intravitreal delivery of AAV-based vectors is mostly safe and well-tolerated [8]. Nonetheless, clinical assessment of patients in current optogenetic clinical trials uses a standardized assessment of ocular inflammation according to the international guidelines of the Standardization of Uveitis Nomenclature Working Group [8,85,86].

## 6. Current Clinical Trials

Although optogenetics is a relatively new technology, there are currently four companies that have advanced optogenetic retinal gene therapies into clinical trials (Table 1): (1) GenSight Biologics, GS030 (Paris, France), (2) Allergan, RST-001 (Dublin, Ireland), (3) Bionic Sight, BS01 (New York, NY, USA), and (4) Nanoscope Therapeutics, vMCO-010 (Bedford, TX, USA).

PIONEER is a Phase 1/2a, open-label, dose-escalation study that was designed to evaluate the safety and tolerability of GS030 in patients with advanced non-syndromic RP (ClinicalTrials.gov, accessed on 1 January 2022 Identifier: NCT03326336). GS030, developed by GenSight Biologics, is a novel optogenetic treatment that combines a drug product (GS030-DP) administered by a single intravitreal injection with a wearable visual stimulation medical device (GS030-MD). GS030-DP introduces a gene encoding the light-sensitive ChR protein ChrimsonR-tdTomato (ChrR-tdT) into RGCs by intravitreal delivery of an AAV2.7m8 capsid variant that features a peptide on its heparin binding site [8,87]. ChrimsonR is a microbial opsin with a peak wavelength sensitivity of 590 nm, which is approximately 100 nm more red-shifted than ChR2. Theoretically, it allows greater safety and causes less pupillary constriction than the highly phototoxic blue light used to activate many other sensors [35,77]. In 2017, GS030-MD was granted Orphan Drug Designation by the FDA for the treatment of RP. The optronic device (GS030-MD) works in conjunction with the gene therapy to activate the newly photosensitive RGCs with local 595 nm light pulses from autonomous pixels that detect light intensity changes in the visual field as distinct events [88]. In May 2021, a case report described the first successful example of partial functional vision recovery after optogenetic therapy in a 58-year-old male patient in the PIONEER study treated with 5.0 × 10^10^ vector genomes (vg) of GS030-DP. Preliminary safety data of the ongoing PIONEER clinical trial suggests good tolerance of GS030-DP in three dose-escalation cohorts administered 5 × 10^10^, 1.5 × 10^11^, and 5.0 × 10^11^ vg per eye (gensight-biologics.com, accessed on 9 September 2021). However, one-year post-injection results of all treated patients are not expected until 2023.

In 2015, Allergan (formerly RetroSense Therapeutics) initiated a Phase 1/2a, open-label, dose-escalation study to evaluate the safety and tolerability of RST-001 in patients with advanced RP (ClinicalTrials.gov, accessed on 1 January 2022 Identifier: NCT02556736). RST-001 is an AAV-2-based vector encoding ChR2 for the transduction of RGCs through intravitreal injection. In June 2021, results from the Phase 1 sequential dose-escalation study reported no serious adverse events, suggesting that RST-001 is well tolerated. However, no data regarding the effect of RST-001 on improving visual function have been released. Because ChR2 is a Type 1 opsin that requires a very high-intensity blue light for activation, this therapy would require an artificial light source that can potentially be toxic to the retina [9]. In 2014, the FDA granted Orphan Drug Designation for RST-001 as treatment for RP, and Allergan plans to expand the indication of RST-001 to include dry AMD. Phase 2a of the study, in which patients will receive RST-001 at the maximum tolerated dose, is currently ongoing.

Bionic Sight has created an enhanced light-sensitive ChR (ChronosFP) that is currently being tested in a Phase 1/2 clinical trial (ClinicalTrials.gov, accessed on 1 January 2022 Identifier: NCT04278131). The open-label, dose-escalation study delivers its BS01 gene therapy to the RGCs of patients with advanced RP by intravitreal injection of a recombinant AAV-2 vector expressing ChronosFP. This study is not only of interest because of its use of a variant of ChR but also because it combines gene therapy with an advanced neuroprosthetic that incorporates the “retina’s neural code” [89]. The prosthetic system consists of two parts: an encoder and a transducer. The encoder uses the retina’s neural code to convert visual input into signals the brain can interpret. The transducer then projects the encoded pulses onto the retina, which creates the potential for patterns of light that are visually meaningful. In March 2021, Bionic Sight announced that the first four patients who received BS01 could detect light and motion (bionicsightllc.com, accessed on 9 September 2021). These early observations are very encouraging, and the company plans to escalate to higher doses of BS01 and report additional results later this year.

Nanoscope Therapeutics is a clinical-stage biopharmaceutical company that has developed an optogenetic therapy employing proprietary ambient light-sensitive, polychromatic opsins that have the potential to restore vision in different color environments without the need for artificial light interventions. RESTORE is an ongoing Phase 2b randomized, placebo-controlled, dose-ranging study that will evaluate the safety and efficacy of a single intravitreal injection of a virus carrying multi-characteristic opsin (vMCO-010) in adults with RP (ClinicalTrials.gov, accessed on 1 January 2022 Identifier: NCT04945772). At the American Academy of Ophthalmology’s 2020 annual meeting, Nanoscope Therapeutics announced preliminary data from their completed Phase 1/2a open-label, dose-escalation study. The company reported that all 11 patients with advanced RP who received a low dose (1.75 × 10^11^ vg per eye) or high dose (3.5 × 10^11^ vg per eye) of vMCO-010 experienced significant dose-dependent improvements in visual acuity from baseline to 16 weeks (nanostherapeutics.com, accessed on 9 September 2021). vMCO-010 uses an AAV2 vector to target bipolar cells, rather than RGCs, which may be preferable in the earlier stages of the disease to preserve as much of the visual processing circuitry as possible [90]. MCO1 is a microbial opsin with a broad activation spectrum that can be stimulated by ambient light ranging from blue to red wavelengths [91]. vMCO-010 has received Orphan Drug Designations for RP and Stargardt macular degeneration from the FDA. In July 2021, the company announced that the first patient was dosed with vMCO-010 in their Phase 2b clinical trial.

## 7. Future Perspectives

In no field have novel experimental therapies, especially those based on genetic engineering, played a more prominent role over the past decade in driving scientific innovation than for the development of innovative treatments in eye disease. Future therapies affecting eye function are likely to benefit from a combination of these novel approaches, including interventions to delay photoreceptor degeneration and more recently conceived methods such as implantation of stem cell/photoreceptor precursors combined with synergistic optogenetic procedures to indirectly or directly restore or completely replace RGC function if the retinal cycle regenerating the photopigment is completely lost.

Future clinical trials in optogenetics are likely to focus on determining the optimal route of delivery of gene therapy without causing inflammation or potentially detaching the retina. Clinical trials for gene therapy to treat LCA have demonstrated the safety and efficacy of subretinal delivery of transgenes to the retina via an AAV2 vector [92,93]. An advantage of subretinal injection is a more favorable biodistribution profile compared to intravitreal injection, thereby decreasing the risk of systemic exposure and inflammation [94]. However, subretinal delivery involves transient detachment of the retina, which may further damage a fragile and degenerated retina. Intravitreal injection is a less technically challenging procedure and therefore less prone to complications [95,96]. Still, higher viral titers are required to reach therapeutic levels of viral transduction from the vitreous, increasing the risk of intraocular and systemic inflammation [82,94,97]. Alternative drug delivery methods, including suprachoroidal injection and sub-internal limiting membrane, are under pre-clinical investigation [98,99].

## Figures and Tables

**Figure 1 biomolecules-12-00269-f001:**
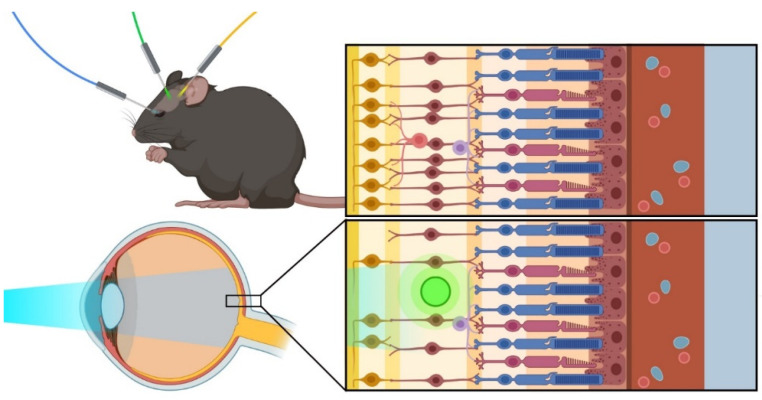
Optogenetic approaches: The favorite models for optogenetic therapy include mouse models, which can be adapted to study and treat different eye-related conditions using various optogenetic techniques.

**Table 1 biomolecules-12-00269-t001:** Summary of ongoing clinical trials of optogenetic therapy for vision restoration.

Company	Disease	Intervention	Opsin	Viral Vector	Delivery Route	Trial Stage	Clinical Trial Identifier
GenSight Biologics	Retinitis Pigmentosa	Drug: GS030-DP Medical device: GS030-MD	ChrimsonR	rAAV2.7m8	Intravitreal injection	Phase 1/2a	NCT03326336
Allergan	Advanced Retinitis Pigmentosa	Drug: RST-001	ChR2	rAAV2.7m8	Intravitreal injection	Phase 1/2a	NCT02556736
Bionic Sight LLC	Retinitis Pigmentosa	Drug: BS01	ChronosFP	rAAV2	Intravitreal injection	Phase 1/2	NCT04278131
Nanoscope Therapeutics Inc.	Retinitis Pigmentosa	Drug: vMCO-010	MCO1	rAAV2	Intravitreal injection	Phase 2b	NCT04945772

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
