# Peer review of "Advances in Ophthalmic Optogenetics: Approaches and Applications"

_biomolecules, 2022, doi:10.3390/biom12020269_

Round 1
Reviewer 1 Report
The review is interesting and a good composite of the current ontogenetic treatments and uses. I mainly had issues with references (not enough of them) and some grammar issues.
Line 19: Too many commas. Try Such approaches are attractive because they are agnostic...
Lines 45 - 47: References needed
Line 57: rely should be relies
Lines 62 - 67: References needed
Lines 73 and 76: Be sure to italicize Latin terms cis and trans
Line 93: You already defined ChR2 and Channelrhodopsin-2 so be sure to not redefine your abbreviations.
Lines 100 - 103: References needed (different variants of ChR)
Line 82: Define human rhodopsin as RHO
Lines 91, 104, 106: Change rhodopsin to RHO after dining it at line 82
Line 115: Change melanopsin to OPN4 as you already defined it as such toward the beginning. Stay consistent.
Line 116: Be careful with this statement "ganglion cells that are not essential for vision" when referring to ipRGCs. This is not 100% true. They are ganglion cells and losing them would affect other cell populations. So even if they are primarily projecting to the SCN for circadian rhythm, they are still essential for the retina to function. Maybe something like "OPN4 is a mammalian opsin expressed in a subpopulation of ganglion cells which serve a more specialized visual function (circadian rhythm entrainment) and are categorized as intrinsically photosensitive retinal ganglion cells (ipRGCs)."
Line 139: CIB1 is not defined.
Line 185: name the VEGF inhibitor gene
Lines 201 - 205: References needed
Lines 212 - 219: References needed
Reviewer 2 Report
In this manuscript, Prosseda et.al, reviews the advances in ophthalmic optogenetics by summarising the optogenetic molecules, vectors, and applications of optogenetics for the treatment of retinal degeneration and glaucoma. This review is a valuable addition to the field as this review summarizes the existing literature on optogenetics, clinical trials and presents the advantages of optogenetics. The article has interesting observations which are beneficial to researchers in the areas of retinal degeneration and other retinal associated complications. However, the manuscript needs to be updated with more details and appropriate original references in several sections. Some major and minor concerns discussed below.
-Line 8: Affiliation 3 is not added to any of the authors.
-Line 58-67: Add more background on the function of photoreceptors, pathogenesis of retinal degeneration and advantages of treatment with optogenetics. Also add appropriate references for this paragraph.
-Line 144: Explain further on LOV-domains with appropriate references.
-Line 195: In this section, diagram/table or further description about summary of in vivo optogenetic tools currently used will be beneficial to the readers.
-Line 195: Also in this section discuss about the clinical examinations and safety parameters used to evaluate optogenetic studies.
-Line 348 - Discuss further on considerations which could advance optogenetic studies in this section.
Round 2
Reviewer 2 Report
Prosseda et.al., have addressed all the concerns raised in the initial manuscript and the efforts are greatly appreciated. The revised review article can deserve publication without any further changes.